# Cosmos Policy: Fine-Tuning Video Models for Visuomotor Control and Planning

**Moo Jin Kim**[1,2]   **Yihuai Gao**[1,2]   **Tsung-Yi Lin**[1]   **Yen-Chen Lin**[1]   **Yunhao Ge**[1]   **Grace Lam**[1]

**Percy Liang**[2]   **Shuran Song**[1,2]   **Ming-Yu Liu**[1]   **Chelsea Finn**[2]   **Jinwei Gu**[1]

[1]NVIDIA   [2]Stanford University

https://research.nvidia.com/labs/dir/cosmos-policy/

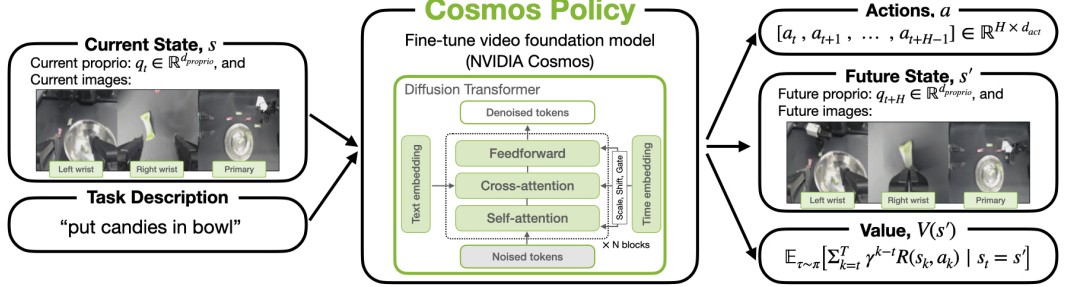

**Figure 1:** We present Cosmos Policy, a state-of-the-art robot policy fine-tuned from the NVIDIA Cosmos-Predict2-2B video foundation model. Cosmos Policy handles multimodal inputs and multi-view camera images and predicts (1) a robot action chunk, (2) future state (represented by robot proprioception and image observations), and (3) value (expected rewards-to-go at the future state). No architectural changes are made to the base video model, and all modalities are jointly modeled through the video diffusion learning objective.

## Abstract

Recent video generation models demonstrate remarkable ability to capture complex physical interactions and scene evolution over time. To leverage their spatiotemporal priors, robotics works have adapted video models for policy learning but introduce complexity by requiring multiple stages of post-training and new architectural components for action generation. In this work, we introduce Cosmos Policy, a simple approach for adapting a large pretrained video model (Cosmos-Predict2) into an effective robot policy through a single stage of post-training on the robot demonstration data collected on the target platform, with no architectural modifications. Cosmos Policy learns to directly generate robot actions encoded as latent frames within the video model's latent diffusion process, harnessing the model's pretrained priors and core learning algorithm to capture complex action distributions. Additionally, Cosmos Policy generates future state images and values (expected cumulative rewards), which are similarly encoded as latent frames, enabling test-time planning of action trajectories with higher likelihood of success. In our evaluations, Cosmos Policy achieves state-of-the-art performance on the LIBERO and RoboCasa simulation benchmarks (98.5% and 67.1% average success rates, respectively) and the highest average score in challenging real-world bimanual manipulation tasks, outperforming strong diffusion policies trained from scratch, video model-based policies, and state-of-the-art vision-language-action models fine-tuned on the same robot demonstrations. Furthermore, given policy rollout data, Cosmos Policy can learn from experience to refine its world model and value function and leverage model-based planning to achieve even higher success rates in challenging tasks. We release code, models, and training data at https://research.nvidia.com/labs/dir/cosmos-policy/.

## 1 Introduction

---

Correspondence to: Moo Jin Kim <moojink@cs.stanford.edu>.

Large pretrained video generation models have shown impressive ability to generate physically plausible and temporally coherent videos (NVIDIA et al., 2025; Wan et al., 2025; Yang et al., 2024; Bao et al., 2024; Kong et al., 2024; Zheng et al., 2024). Unlike pretrained vision-language models—which learn semantic concepts from static image-text pairs and have been popularized as robot policy backbones by recent vision-language-action (VLA) model research ((Brohan et al., 2023; Kim et al., 2024; Intelligence et al., 2025; Li et al., 2025b))—pretrained video generation models learn temporal causality, implicit physics, and motion patterns from millions of videos. These spatiotemporal priors hold significant value for robotics applications. In this work, we explore how to effectively leverage video models for robotic control and how they can incorporate policy rollout data to refine their world models and enable more effective planning.

Prior works have made significant progress on adapting video models for robotic manipulation, leveraging both robot action data and "action-less" Internet video data to train generalizable policies and perform new tasks with small amounts of demonstrations (Liang et al., 2025; Zhong et al., 2025; Hu et al., 2024; Liao et al., 2025; Unitree, 2025; Feng et al., 2025; Yang et al., 2025; Wang et al., 2025). However, these works often require multiple training stages (e.g., video fine-tuning followed by action module training) and introduce new architectural components, such as separate action diffusers or inverse dynamics models. Other works avoid these complexities by training unified video-action models (Li et al., 2025a; Zhu et al., 2025), but they do not leverage pretrained video models due to their custom design, limiting their ability to capitalize on the spatiotemporal priors.

In this work, we address these limitations with **Cosmos Policy**: an effective robot policy that is adapted from a pretrained video model (Cosmos-Predict2-2B (NVIDIA et al., 2025)) through a single stage of post-training on robot demonstrations. Unlike prior works which carefully design separate action modules and algorithms, Cosmos Policy makes no architectural modifications and instead leverages the pretrained model's core learning mechanism to capture action distributions. Since video models are effective at modeling complex, high-dimensional, multimodal distributions and can generate temporally coherent videos with hundreds of frames, we hypothesize that their learning algorithms are well-suited for representing actions alongside other modalities. Following this reasoning, we directly fine-tune a video model to simultaneously generate robot actions, future state images, and future state values (expected total cumulative rewards), all of which we encode as latent frames within the model's latent diffusion sequence. With future state and value predictions, Cosmos Policy can use best-of-N sampling to plan by generating candidate actions, imagining their resulting future states, ranking these states by predicted value, and executing the highest-value action. This search process produces trajectories that are more likely to succeed at the task

Our main contribution is the Cosmos Policy approach for fine-tuning pretrained video models to incorporate different modalities that enable visuomotor control and planning. We evaluate our method in two modes: first as a direct policy (without planning) and then with model-based planning using the future state and value predictions. As a direct policy, Cosmos Policy achieves a new state of the art in both the LIBERO and RoboCasa simulation benchmarks (98.5% and 67.1% average success rates, respectively), outperforming diffusion-based policies trained from scratch, video-based policies (e.g., UVA, Video Policy), and even fine-tuned VLAs (e.g., $\pi_{0.5}$, OpenVLA-OFT, CogVLA, UniVLA, DP-VLA, GR00T-N1.5). It also achieves the highest average success rate (93.6%) among state-of-the-art policies in challenging real-world bimanual manipulation tasks. Further, when enhanced with model-based planning, we observe a 12.5 percent higher task completion rate on average in two challenging real-world manipulation tasks. In these experiments, we show that Cosmos Policy can incorporate past experiences from policy rollouts to refine its world model and value function and plan more effectively. Lastly, we compare our model-based planning approach to a model-free variant and study their relative advantages.

## 2 RELATED WORK

**Video-based robot policies.** Recent works have made great strides in leveraging video models for manipulation. Some methods first fine-tune video models on robot data and then train separate action modules to predict robot actions from generated video frames (Liang et al., 2025; Zhong et al., 2025; Hu et al., 2024; Liao et al., 2025; Unitree, 2025; Feng et al., 2025; Yang et al., 2025; Wang et al., 2025; He et al., 2024). Other works train unified video-action models that jointly predict future frames and actions (Li et al., 2025a; Zhu et al., 2025), but these approaches do not leverage pretrained video models and thus do not benefit from their spatiotemporal priors. In contrast to these

works, we propose a single-stage fine-tuning approach that directly adapts pretrained video models to generate actions (as well as other modalities such as robot proprioceptive state and state values) within their native latent diffusion process.

**Vision-language-action models.** State-of-the-art robotic manipulation policies increasingly leverage large pretrained backbones. Vision-language-action (VLA) models such as RT-2 (Brohan et al., 2023), OpenVLA (Kim et al., 2024), $\pi_{0.5}$ (Intelligence et al., 2025), UniVLA (Bu et al., 2025), and CogVLA (Li et al., 2025b) fine-tune vision-language models on large-scale robotic imitation data, achieving strong performance across diverse manipulation tasks. While these methods exhibit strong generalization to various semantic concepts unseen in robotic interaction data, they leverage pretrained models that have mostly been trained on static image-text pairs rather than videos. In contrast to these VLAs, we leverage a pretrained video model that has learned spatiotemporal dynamics and implicit physics from predicting future frames for Internet-scale datasets. We hypothesize that this different pretrained backbone can serve as a strong foundation for low-level control policies.

**World models and value functions.** World models have been used in various ways in robotics and reinforcement learning, from classical model-predictive control to modern neural approaches. Influential works such as Dyna (Sutton, 1991), MBPO (Janner et al., 2019), TD-MPC (Hansen et al., 2022; 2023), and the Dreamer family of works (Hafner et al., 2019; 2020; 2023) demonstrate the benefits of integrating planning with learning, using learned dynamics models to improve decision making in various control tasks. Recent works have explored different paradigms: FLARE (Zheng et al., 2025) adds learnable future tokens to diffusion transformer sequences to predict compact representations of future state, SAILOR (Jain et al., 2025) uses separate world and reward models with MPPI planning to iteratively search for better actions and refine the base policy, and Latent Policy Steering (Wang et al., 2025) pretrains world models using optical flow as an embodiment-agnostic action representation and subsequently trains a separate value function to steer the policy towards states with higher rewards. In contrast to these prior works that rely on separate modules for the policy, world model, and value function and typically train from models from scratch, we use a single unified architecture that serves simultaneously as the policy, world model, and value function and initialize from a pretrained video model.

## 3 PRELIMINARIES

**Cosmos video model.** The pretrained video model that serves as the initialization for Cosmos Policy is Cosmos-Predict2-2B-Video2World (NVIDIA et al., 2025), a latent video diffusion model that receives a starting image and textual description as input and predicts subsequent frames to create a short video. The model operates over continuous tokens encoded by the Wan2.1 spatiotemporal VAE tokenizer (Wan et al., 2025) and is trained using the EDM denoising score matching formulation (Karras et al., 2022). The core training objective for the denoiser network $D_\theta$ at noise level $\sigma$ is: $\mathcal{L}(D_\theta, \sigma) = \mathbb{E}_{\mathbf{x}_0, \mathbf{c}, \mathbf{n}} \left[ \| D_\theta(\mathbf{x}_0 + \mathbf{n}; \sigma, \mathbf{c}) - \mathbf{x}_0 \|_2^2 \right]$, where $\mathbf{x}_0$ is a clean VAE-encoded image sequence, $\mathbf{c}$ represents the textual description encoded as T5-XXL embeddings (Raffel et al., 2020), $\mathbf{n} \sim \mathcal{N}(\mathbf{0}, \sigma^2 \mathbf{I})$ is i.i.d. Gaussian noise used to corrupt $\mathbf{x}_0$, and $D_\theta$ is a diffusion transformer (Peebles & Xie, 2023) that learns to recover the clean sample given the corrupted one. $D_\theta$ conditions on $\mathbf{c}$ via cross-attention and on $\sigma$ via adaptive layer normalization (Perez et al., 2018; Peebles & Xie, 2023). The Wan2.1 tokenizer compresses a video sequence of size $(1 + T) \times H \times W \times 3$ into a latent sequence of size $(1 + T') \times H' \times W' \times 16$, where $T' = \frac{T}{4}$, $H' = \frac{H}{8}$, $W' = \frac{W}{8}$; these resulting latent frames compose $\mathbf{x}_0$ above. The first frame undergoes no temporal compression to allow for conditioning on a single input image. During training, a conditioning mask is used to ensure that the first latent frame corresponding to the input image remains clean (without noise) while subsequent frames are corrupted with noise.

**MDP formulation and imitation learning.** We frame robotic manipulation tasks as finite-horizon Markov decision processes (MDPs) defined by the tuple $\langle S, A, T, R, H \rangle$, where $S$ is a set of states, $A$ is a set of actions, $T : S \times A \rightarrow \Pi(S)$ is the state transition function, $R : S \times A \rightarrow \mathbb{R}$ is the reward function, and $H \in \mathbb{N}$ is the time horizon, with time steps $t \in \{1, 2, \ldots, H\}$. We train a policy $\pi : S \rightarrow \Pi(A)$ to maximize rewards, using sparse rewards where $R(s_t, a_t) = 0$ for $t < H$ and terminal rewards $R(s_H, a_H) \in [0, 1]$. We train policies via imitation learning on expert demonstrations containing state-action pairs. All policies predict action chunks—sequences of actions for multiple timesteps—to improve motion smoothness and success rates.

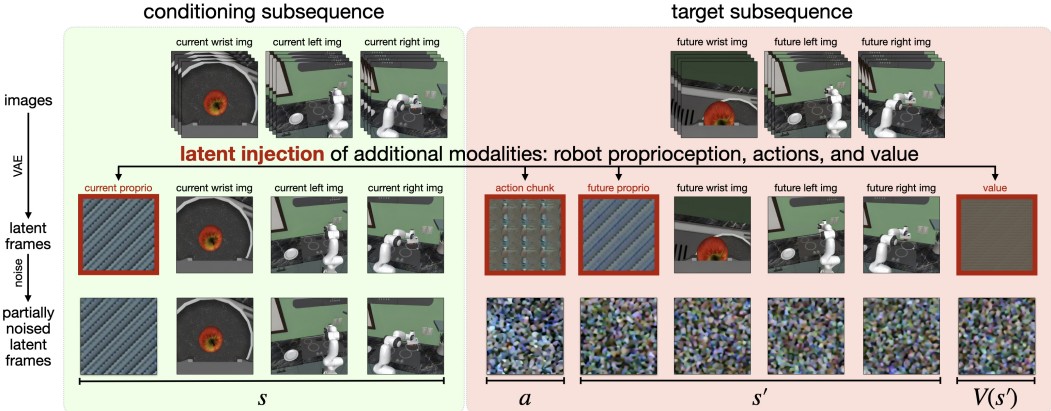

**Figure 2: The latent diffusion sequence of Cosmos Policy.** We illustrate *latent frame injection*—the primary mechanism for adapting the pretrained Cosmos-Predict2 into a policy that can predict robot actions, future states, and values without architectural changes. First, raw images are tokenized into latent frames (first row). Then, additional modalities are inserted directly into the latent frame sequence of the video diffusion model (second row). The model is then tasked to denoise the noised latent frames conditioned on the clean frames (third row). See Section 4.1 for more details. (Note: For simplicity, this figure does not depict certain implementation details; see Figure 8 for a more detailed visualization.)

**World models and value functions.** A *world model* $\hat{T} : S \times A \to \Pi(S)$ learns to predict the future state given current state and action, approximating the true environment dynamics. The *value function* for a policy $\pi$ at state $s$ represents expected discounted returns from $s$ under $\pi$. It is defined as $V^\pi(s) = \mathbb{E}_{\tau \sim \pi} \left[ \sum_{k=t}^{H} \gamma^{k-t} R(s_k, a_k) \mid s_t = s \right] = \mathbb{E}_{\tau \sim \pi} \left[ \gamma^{H-t} R(s_H, a_H) \mid s_t = s \right]$ in the sparse reward setting, where $\gamma$ is a discount factor that backpropagates the terminal reward through time. We simply use Monte Carlo returns in this work, labeling each transition in a rollout with the observed return $\gamma^{H-t} R(s_H, a_H)$. Note: To be precise, we acknowledge that the true state is not fully observable, and the world model predicts future *observations* (robot proprioception and camera images). However, for notational simplicity and readability, we opt to use the term "state" and treat observations as approximations of the state.

# 4  COSMOS POLICY: ADAPTING VIDEO MODEL FOR CONTROL & PLANNING

In this section, we discuss how to adapt Cosmos-Predict2 into a unified model that predicts actions, future states, and values. We also discuss leveraging policy rollout data to enable effective planning.

## 4.1  LATENT FRAME INJECTION: INCORPORATING NEW MODALITIES

The original Cosmos-Predict2 model takes as input an image and a textual description to generate a short video for a single camera view. It does not support robot proprioception as input, robot actions or state values as output, nor multiple camera views—all of which are desired or required for manipulation policies.

Rather than designing new model components or making architectural modifications as done in prior works, we propose to encode additional modalities as new latent frames that are directly injected into the video model's latent diffusion sequence. Given a $(1 + T') \times H' \times W' \times 16$ sequence of latent frames, which originally correspond to images in a video, we interleave new modalities (robot state, action chunk, and state values) and images from additional camera views by inserting new latent frames between existing image latent frames. For multiple camera viewpoints, the process is simpler: we simply insert the additional camera images at the image sequence level as shown in the top row of Figure 2 (and the model is subsequently fine-tuned to handle these new viewpoints).

We now discuss an illustrative example of latent injection for incorporating new modalities. For a robotic platform with two static third-person cameras and a wrist-mounted camera, our latent

sequence contains 11 latent frames: (1) a blank placeholder,[*] (2) robot proprioception (e.g., end-effector pose or joint angles), (3) wrist camera image, (4) first third-person camera image, (5) second third-person camera image, (6) action chunk, (7) future robot proprioception, (8) future wrist camera image, (9) future first third-person camera image, (10) future second third-person camera image, and (11) future state value. Among these, (2), (6), (7), and (11) represent new modalities while (3), (5), (8), and (10) represent additional camera views (assuming that the first third-person camera is the "primary" camera). To encode the new modalities as latent frames, we fill each $H' \times W' \times C'$ latent volume with normalized and duplicated copies of the robot proprioception, action chunk, or value (where normalization simply consists of rescaling to $[-1, +1]$). See Figure 2 for an illustration. This ordering of modalities in the sequence represents $(s, a, s', V(s'))$, and it allows for autoregressive decoding of actions, future state, and future state value from left to right (see Section 4.2 for further discussions on this). Note that $s$ and $s'$ only consist of the observations at time $t$ and $t + K$, respectively, where $K$ is the action chunk size. In other words, we do not use input history nor predict future frames across multiple subsequent timesteps. Lastly, latent injection is flexible and can be adapted for any particular robot setup: for example, for a robot with only one third-person camera, one can remove the latent frames corresponding to additional camera viewpoints, and this would result in only seven total latent frames. See latent injection implementation details in Appendix A.1.

## 4.2 Joint Training of Policy, World Model, & Value Function

**Implementing joint training objectives.** Now that we have a latent diffusion scheme that incorporates additional modalities and camera views that are compatible with robotic policy learning, we can adapt the video model into a policy by training on robot data. For each training step, we sample a batch of $(s, a, s', V(s'))$ tuples.[†] 50 percent of the batch is sampled from the *demonstrations dataset* and is used to train the policy ($p(a, s', V(s')|s)$), while the other 50 percent is sampled from the *rollouts dataset* and is split into two halves: one half for training the world model ($p(s', V(s')|s, a)$) and the other half for training the value function ($p(V(s')|s, a, s')$). The conditioning scheme—i.e., which part of the latent diffusion sequence is used as conditioning and which part is used as the target to generate—determines which of these three functions is being trained (see Figure 12 for more details). Initially, the rollouts dataset is simply a superset of the demonstrations dataset that also includes failed demonstrations, if they exist. (Failed demonstrations are those that do not successfully complete the task when replayed in the environment due to human error during data collection, e.g., in the LIBERO and RoboCasa simulations where roughly 10 to 20 percent of demonstrations fail when replayed. In certain environments where teleoperation data is collected more carefully, such as our real-world ALOHA environment, failed demonstrations do not exist; in this case, the demonstrations dataset and rollouts dataset are equal.)

Note that policy and world model training involves auxiliary targets, i.e., the policy is trained to model not just $p(a|s)$ but rather $p(a, s', V(s')|s)$, and the world model learns not just $p(s'|s, a)$ but rather $p(s', V(s')|s, a)$. We find in Section 5.2 that the auxiliary supervision improves policy performance. Also, note that the $V(s')$ predictions are conditioned on the full latent prefix (i.e., all of $(s, a, s')$) during initial Cosmos Policy training. However, when we later fine-tune this base checkpoint on policy rollout data to produce a model with more accurate future state and value predictions, we can choose to condition the value generation on a subset of $(s, a, s')$ via input masking. The choice of the input mask determines whether the value function represents the state value $V(s')$ or state-action value $Q(s, a)$; we compare these variations in planning experiments (Section 5.3).

**Parallel vs. autoregressive decoding.** Since Cosmos Policy learns to both jointly and conditionally predict the targets $(a, s', V(s'))$ based on apportioned training samples, it can generate actions, future states, and values either jointly in parallel or autoregressively from left to right. Parallel decoding offers greater speed, while autoregressive decoding provides higher-quality predictions and allow for separate checkpoints to be used for the policy versus the world model and value function. For direct policy evaluation without planning, only actions are required for task execution while the latter two outputs can be discarded, so we use parallel decoding in this case. For evaluations with planning, we enable autoregressive decoding for higher-quality future state and value predictions.

---

[*]This is an implementation detail that is necessary due to the VAE's $(1 + \frac{T}{4})$ temporal compression scheme discussed in Section 3, which is commonly used in models such as Cosmos-Predict2 and Wan2.1. See Appendix A.1 for details.

[†]We technically sample the empirical return $G_t = \gamma^{H-t} R(s_H, a_H)$ but write "$V(s')$" for notational simplicity and readability.

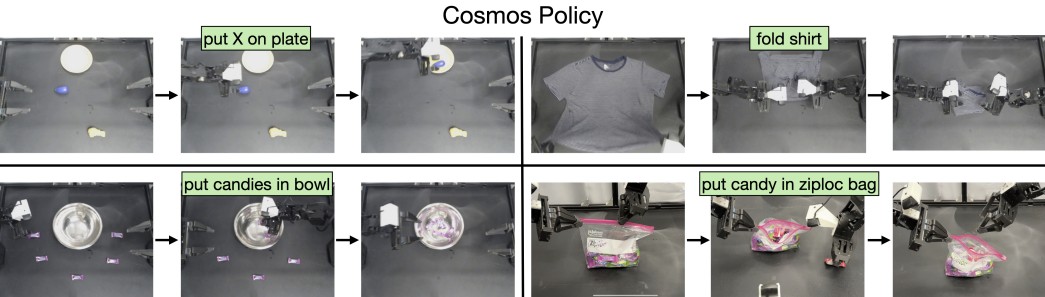

**Figure 3: Cosmos Policy in the ALOHA robot tasks.** Cosmos Policy can successfully execute real-world robotic control tasks that require long-horizon, high-precision manipulation and have high action multimodality.

### 4.3 PLANNING WITH COSMOS POLICY'S WORLD MODEL AND VALUE FUNCTION

Cosmos Policy can be deployed as (1) a direct policy without planning or (2) a planning policy using future state and value predictions to search for higher-quality actions. However, training on demonstrations alone is insufficient for effective planning since the data only covers successful outcomes,[‡] which means that the world model and value function see a narrow state-action distribution and may struggle to generalize beyond that distribution. We thus find it critical to collect policy rollout data and learn from these experiences.

**Learning from rollouts.** We collect rollout data by deploying Cosmos Policy in diverse initial conditions and recording the trajectory and episode outcome (success/fail or a fractional score). Given the rollout dataset, we fine-tune our Cosmos Policy checkpoint with heavier weighting on the world model and value function predictions: 90 percent of each training batch is split evenly world model and value function training, and only 10 percent is used to train the policy.

Once we have the fine-tuned checkpoint for refined world modeling and policy learning, we propose *dual deployment*: the original Cosmos Policy checkpoint serves as the policy (we thus call it the "policy model"), while the refined checkpoint serves as the world model and value function (we thus call it the "planning model"). This ensures that the refined world model and value function are trained on on-policy data collected by the original policy.

**Model-based planning.** Given the policy model and the planning model, we implement best-of-N sampling as follows: (1) sample multiple action proposals from the policy, (2) use the planning model to predict the future state and value for each proposal, (3) select and deploy the action that leads to the predicted state with the highest predicted value. For greater accuracy and better modeling of potentially multimodal future state and value distributions, we ensemble the predictions by querying the world model three times per action and the value function five times per future state, resulting in fifteen total value predictions for each action proposal. We aggregate these via "majority mean": we determine whether the majority predict success or failure (via a fixed threshold) and then average values within the majority group. This approach is more robust to outliers than naive averaging when value predictions are bimodal or exhibit high variance.

To speed up the search process, we use parallelized inference, using N GPUs in best-of-N sampling. We also execute the full action chunk (rather only part of it, as done in receding-horizon control) to avoid further increases in computational cost.

## 5 EXPERIMENTS

We evaluate Cosmos Policy to answer four key questions: **(Q1)** How does Cosmos Policy compare with state-of-the-art imitation learning policies when used as a direct policy? **(Q2)** How important are different components of Cosmos Policy? **(Q3)** Can Cosmos Policy leverage rollout experiences and learn an accurate world model and value function for effective planning? **(Q4)** Is it more effective to search using a world model and state value function or a Q-value function (a model-free

---

[‡]Some demonstration datasets, including the LIBERO simulation benchmark training set, includes suboptimal behaviors that may not lead to success when replayed in the training environment, due to human errors during teleoperation. Typically, however, most (if not all) trajectories in a demonstration dataset used for imitation learning are successful.

variation)? We answer these through simulated and real-world evaluations spanning single-arm and dual-arm manipulation tasks.

## 5.1 EXPERIMENTAL SETUP

We now describe the three task suites used in our evaluations. Note that further training and evaluation details are available in Appendices A.2 and A.3.

**LIBERO simulation benchmark.** The LIBERO benchmark (Liu et al., 2024) consists of a variety of environments and tasks featuring a single Franka Emika Panda robot arm. The four primary task suites include LIBERO-Spatial, LIBERO-Object, LIBERO-Goal, and LIBERO-Long (also called LIBERO-10); these assess a policy's ability to handle different spatial layouts, objects, language-specified goals, and long-horizon tasks, respectively. Each task suite provides a training dataset of 500 total demonstrations (10 tasks and 50 demonstrations each). Following Kim et al. (2024), we filter unsuccessful demonstrations for policy training but use the full unfiltered set for world model and value function training.

**RoboCasa simulation benchmark.** The RoboCasa benchmark (Nasiriany et al., 2024) consists of 24 static kitchen manipulation tasks featuring a single Franka Emika Panda robot arm. We follow the evaluation protocol of several prior works (Nasiriany et al., 2024; Bjorck et al., 2025; Zheng et al., 2025; Han et al., 2024; Jang et al., 2025; Liang et al., 2025). Specifically, for each task, success rate is evaluated over 50 trials across five evaluation scenes with different floor plans and styles (10 trials per scene), and the average success rate is computed across all 24 tasks over 3 random seeds (3600 trials total). Unlike LIBERO evaluations, the RoboCasa evaluations only consist of *unseen object instances*, and two of the five scenes per task include styles *never encountered in the training data*.

The benchmark provides a set of 50 human-teleoperated demonstrations for each task and an additional set of 1000 demonstrations generated via MimicGen (Mandlekar et al., 2023), and prior works have shown clear increases in success rates from using larger training datasets (Nasiriany et al., 2024; Bjorck et al., 2025; Zheng et al., 2025; Liang et al., 2025; Jang et al., 2025). However, to assess the relative data efficiency of Cosmos Policy compared to prior works, we train our method on the 50 human-teleoperated demonstrations alone. Similar to LIBERO data preprocessing, we filter unsuccessful demonstrations for policy training but use the full unfiltered dataset for world model and value function training.

**Real-world ALOHA robot tasks.** The ALOHA platform (Zhao et al., 2023) consists of two ViperX 300 S robot arms with three cameras: one top-down and two wrist-mounted. We reduce the controller frequency from 50 Hz to 25 Hz for computational efficiency. All policies take as input robot proprioceptive state (14 joint angles), three camera images, and task descriptions, predicting action chunks of 50 timesteps (2 seconds).[§] We deploy the full action chunk before requerying the policy.

Our evaluation suite consists of four challenging bimanual manipulation tasks (shown in Figure 3): (1) "put X on plate" (80 demos): place objects on a plate based on language instructions, testing language following; (2) "fold shirt" (15 demos): fold one of three T-shirts in multiple steps, testing long-horizon contact-rich manipulation; (3) "put candies in bowl" (45 demos): collect scattered candies, testing ability to handle multimodal grasp sequences; and (4) "put candy in ziploc bag" (45 demos): open and place items in a ziploc slider bag, testing high-precision manipulation with millimeter tolerance.

The evaluations consist of both in-distribution and out-of-distribution testing conditions, with 101 trials total per method across all tasks. We ensure fair comparison between methods by using the same fixed set of initial states for each method.

## 5.2 COMPARING AGAINST STATE-OF-THE-ART IMITATION POLICIES WITHOUT PLANNING

Here we aim to answer questions **Q1** and **Q2** posed in the beginning of this section. We answer **Q1** by comparing Cosmos Policy as a direct policy (without planning) with state-of-the-art imitation learning policies and assessing their relative effectiveness. We answer **Q2** by ablating various components of Cosmos Policy and analyzing the resulting effects on task performance.

---

[§]Diffusion Policy is an exception, as it predicts 48-timestep action chunks since the implementation requires a multiple of 4.

**Table 1: LIBERO simulation benchmark results.** Success rates (SR) across four LIBERO benchmark task suites (Liu et al., 2024). Cosmos Policy success rates are averaged over 500 trials for each suite (10 tasks × 50 episodes) and three random seeds (6000 trials total). Our method achieves highest performance overall, even outperforming fine-tuned state-of-the-art vision-language-action (VLA) models.

**Table 2: RoboCasa simulation benchmark results.** Success rates (SR) across 24 kitchen manipulation tasks (Nasiriany et al., 2024). Cosmos Policy success rates are averaged over 50 trials for each task and three random seeds (3600 trials total). Our method achieves a state-of-the-art average success rate of 67.1% while requiring significantly fewer training demonstrations (50 versus >300).

| | Spatial SR (%) | Object SR (%) | Goal SR (%) | Long SR (%) | Average SR (%) |
|---|---|---|---|---|---|
| Diffusion Policy (Chi et al., 2023) | 78.3 | 92.5 | 68.3 | 50.5 | 72.4 |
| Dita (Hou et al., 2025) | 97.4 | 94.8 | 93.2 | 83.6 | 92.3 |
| $\pi_0$ (Black et al., 2024) | 96.8 | 98.8 | 95.8 | 85.2 | 94.2 |
| UVA (Li et al., 2025a) | – | – | – | – | 90.0 |
| UniVLA (Bu et al., 2025) | 96.5 | 96.8 | 95.6 | 92.0 | 95.2 |
| $\pi_{0.5}$ (Intelligence et al., 2025) | **98.8** | 98.2 | 98.0 | 92.4 | 96.9 |
| Video Policy (Liang et al., 2025) | – | – | – | – | 94.0 |
| OpenVLA-OFT (Kim et al., 2025) | 97.6 | 98.4 | 97.9 | 94.5 | 97.1 |
| CogVLA (Li et al., 2025b) | 98.6 | 98.8 | 96.6 | 95.4 | 97.4 |
| Cosmos Policy (ours) | 98.1 | **100.0** | **98.2** | **97.6** | **98.5** |

| | # Training Demos per Task | Average SR (%) |
|---|---|---|
| GR00T-N1 (Bjorck et al., 2025) | 300 | 49.6 |
| UVA (Li et al., 2025a) | 50 | 50.0 |
| DP-VLA (Han et al., 2024) | 3000 | 57.3 |
| GR00T-N1 + DreamGen (Jang et al., 2025) | 300 (+ 10000 synthetic) | 57.6 |
| GR00T-N1 + DUST (Won et al., 2025) | 300 | 58.5 |
| UWM (Zhu et al., 2025) | 1000 | 60.8 |
| $\pi_0$ (Black et al., 2024) | 300 | 62.5 |
| GR00T-N1.5 (Bjorck et al., 2025) | 300 | 64.1 |
| Video Policy (Liang et al., 2025) | 300 | 66.0 |
| FLARE (Zheng et al., 2025) | 300 | 66.4 |
| GR00T-N1.5 + HAMLET (Koo et al., 2025) | 300 | 66.4 |
| Cosmos Policy (ours) | 50 | **67.1** |

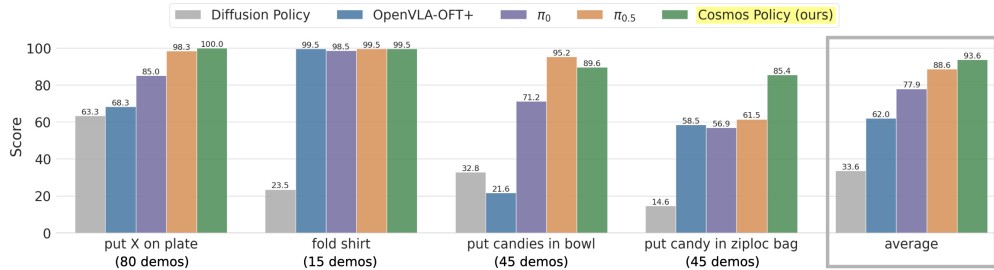

**Figure 4: Real-world ALOHA robot evaluation results.** We evaluate state-of-the-art policies on a suite of four tasks and measure the score, which represents average percent completion of each task. Cosmos Policy achieves highest overall score, outperforming all other methods in three of four tasks.

**Methods in comparison.** In LIBERO and Robocasa, we compare against recent top-performing methods including diffusion-based policies trained from scratch (Diffusion Policy (Chi et al., 2023), Dita (Hou et al., 2025)), video model-based policies (UVA (Li et al., 2025a), UWM (Zhu et al., 2025), Video Policy (Liang et al., 2025)), and fine-tuned VLA models ($\pi_0$, $\pi_{0.5}$, OpenVLA-OFT, CogVLA, UniVLA, DP-VLA (Han et al., 2024), GR00T-N1.5 (Bjorck et al., 2025)). In real-world ALOHA evaluations, we compare against policies that have demonstrated strong performance in real-world bimanual manipulation tasks: Diffusion Policy, OpenVLA-OFT+, $\pi_0$, and $\pi_{0.5}$.

**Results.** Tables 1 and 2 show the performance of Cosmos Policy and prior works in LIBERO and RoboCasa, respectively, while Figure 4 shows performance on the ALOHA robot. We find that Cosmos Policy achieves highest overall performance in all three domains, while establishing a new state of the art in the LIBERO and RoboCasa benchmarks with 98.5% and 67.1% average success rates, respectively. These results demonstrate Cosmos Policy's strong multi-task manipulation performance in both in-distribution and out-of-distribution generalization scenarios. In addition, in ALOHA robot evaluations, we find that Cosmos Policy outperforms fine-tuned VLAs $\pi_{0.5}$ and OpenVLA-OFT+—which have been pretrained on large amounts of robotic imitation data—despite not having benefited from similar large-scale action supervision. This finding suggests that video model priors provide a strong initialization for control policies without requiring additional action-labeled robot data. Sample Cosmos Policy rollouts are visualized in Figure 3.

Qualitatively, we find that while the fine-tuned VLAs show strong performance on the first two tasks, they encounter difficulties in the last two tasks—"put candies in bowl" and "put candy in ziploc bag"—which require handling high action multimodality and executing high-precision grasps, respectively. Figure 5 visualizes two common failure modes of $\pi_{0.5}$ and OpenVLA-OFT+: (1) $\pi_{0.5}$, despite showing highly competitive performance on the first three tasks, struggles to reliably handle the ziploc bag, often missing the initial grasp of the slider with the right arm or not grasping the left side of the bag securely enough with the left arm. (2) OpenVLA-OFT+ often reaches in between two candies rather than directly going for one; we hypothesize that its L1 regression of actions leads to inaccurate modeling of the action distribution in tasks with high multimodality. Compared to

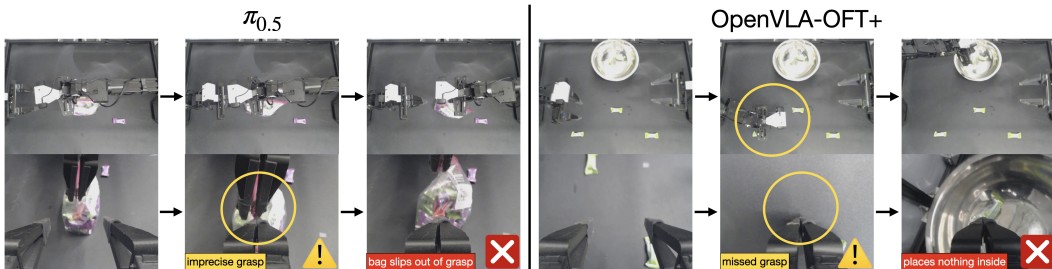

**Figure 5: Common failure modes of $\pi_{0.5}$ and OpenVLA-OFT+ on two challenging ALOHA robot tasks.** **Left:** $\pi_{0.5}$ struggles to execute a high-precision grasp and loses grip of the ziploc bag. **Right:** OpenVLA-OFT+ reaches between two candies rather than towards one, suggesting difficulty with modeling the highly multimodal action distribution.

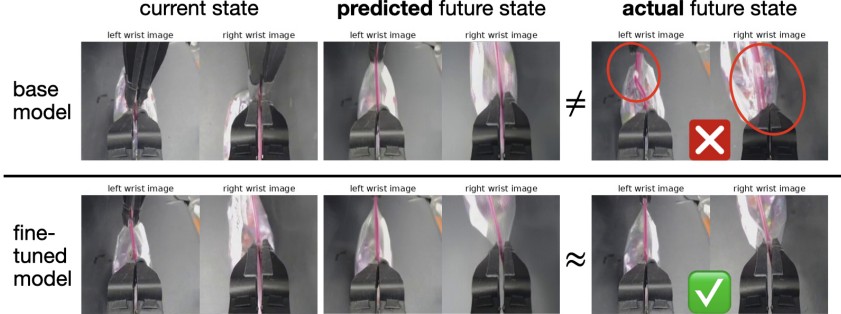

**Figure 6: World model predictions: base Cosmos Policy vs. fine-tuned checkpoint. Top:** The base Cosmos Policy's world model may fail to predict errors such as losing grasp of the ziploc bag slider, as it is only trained on demonstrations. **Bottom:** After fine-tuning on policy rollout data, the world model more accurately predicts the resulting state, enabling more effective planning and eventual episode success.

these methods, Cosmos Policy handles both high multimodality and high precision with substantially greater reliability.

**Ablation experiments.** Recall from Section 4.2 that Cosmos Policy's policy and world model training involves additional targets which provide additional supervision: the policy learns to jointly predict $p(a, s', v(s'))|s)$ instead of $p(a|s)$, and the world model learns to jointly predict $p(s', V(s')|s, a)$ instead of $p(s'|s, a)$. To evaluate the effect of these auxiliary learning objectives, we train a version of Cosmos Policy without them by masking the loss on the additional targets. In addition, we assess the importance of the video model priors by training Cosmos Policy from randomly initialized weights. We use the same number of gradient steps as the full policy for both of these variants. As shown in Table 4, removing the auxiliary losses leads to a 1.5% absolute drop in average success rate while training from scratch leads to a 3.9% drop, suggesting that these components are important for maximal performance. We further evaluate Cosmos Policy trained from scratch on the ALOHA robot for additional supporting evidence and find that it obtains an average score of 80.8 on the "fold shirt" task, which is 18.7 points lower than the full Cosmos Policy. Qualitatively, the from-scratch variant exhibits jerky motions that may damage the robot over prolonged deployment, so we halt further evaluations with it. Additional ablation studies on the Cosmos Policy design and joint training scheme are discussed in Appendix A.4.1.

## 5.3 EVALUATIONS OF COSMOS POLICY WITH MODEL-BASED PLANNING

Here we aim to answer **Q3** by evaluating Cosmos Policy when deployed with model-based planning (as described in Section 4.3), and **Q4** by analyzing how the proposed model-based approach compares different variants of planning, such as directly learning a Q-value function without a world model. Since our base Cosmos Policy already obtains high success rates in LIBERO and on the first two ALOHA robot tasks, we focus our study on the last two more challenging ALOHA robot tasks ("put candies in bowl" and "put candy in ziploc bag"), where there is more room for improvement. Further, we focus on a more challenging set of initial conditions (difficult in-distribution conditions and OOD conditions) and assess whether planning can enhance performance in these settings.

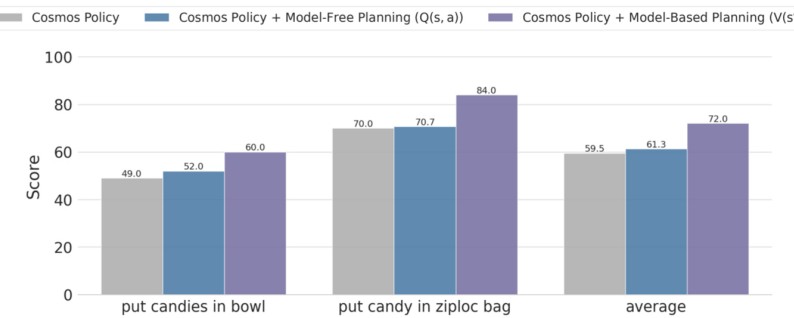

**Figure 7: Model-based planning results.** We evaluate the base Cosmos Policy on challenging initial states for the last two ALOHA robot tasks, and compare it with two planning variants (model-based and model-free). We find that the model-based variant ($V(s')$) leads to highest overall performance.

**Rollout data collection.** To refine Cosmos Policy's world model and value function predictions and enable more effective planning, we gather a rollout dataset that we use for post-training. Conveniently, by running the prior direct policy evaluations, we have already aggregated 505 policy rollouts across all policies. Adding to this, we collect 143 more rollouts from Cosmos Policy for the "put candy in ziploc bag" task. The additional episodes are important for this task since training an accurate world model for it is particularly challenging due to low camera observability from the robot's self-occlusion and highly stochastic environment dynamics where even millimeter differences in control can dictate success or failure. We fine-tune the base Cosmos Policy checkpoint on this pool of 648 rollouts to produce a refined "planning model" for world modeling and value prediction, as described in Section 4.3.

**Comparing different value function formulations.** When fine-tuning the base Cosmos Policy checkpoint on the rollout dataset, we use three independent formulations for value function training by using input masks to condition the value predictions on different subsets of inputs: $V(s')$ (mask out $(s, a)$) or $Q(s, a)$ (mask out $s'$). The $V(s')$ variant requires a world model to predict the future state before the value can be estimated, while the $Q(s, a)$ variant enables model-free planning by directly predicting Q-values without future state predictions.

**Results.** We observe that model-based planning using the $V(s')$ formulation consistently improves success rates over the base Cosmos Policy without planning, as shown in Figure 7. In ALOHA tasks, we observe a 12.5-point average score increase in the two challenging manipulation tasks which involve multimodal grasp sequences and high-precision manipulation. This is a notable improvement given the limited amount of rollout data available for refining the planning model. Qualitatively, we find that the fine-tuned planning model predicts future states more accurately (see Figure 6) and can plan more effectively, ultimately avoiding making mistakes that the base Cosmos Policy makes, such as losing grasp of the slider while opening the ziploc bag. When comparing model-based ($V(s')$) versus model-free ($Q(s, a)$) planning variants, we observe higher performance with the former, which we attribute its ability to leverage learned environment dynamics for more effective and sample-efficient planning. Given a limited amount of rollout data, we expect difficulty with learning an accurate Q-function and suspect that the model may also overfit given higher input dimensionality.

# 6 DISCUSSION

We presented Cosmos Policy, a state-of-the-art robot policy fine-tuned from the Cosmos-Predict2 video foundation model that demonstrates strong performance in LIBERO, RoboCasa, and ALOHA robot environments. We also show that incorporating policy rollout data to refine world model and value function predictions enables effective model-based planning. **Limitations and future work:** We observe substantially lower inference speed when using model-based planning (e.g., around 5 seconds to produce one action chunk), which may limit applicability to dynamic tasks. How to speed up the search is an important direction for future study. In addition, effective planning requires substantial rollout data to achieve accurate predictions beyond the demonstration distribution. Learning from fewer rollouts would increase the accessibility of our approach. Lastly, we focus on best-of-N planning with one layer in the search tree; extending the world model's prediction horizon and planning to greater depths could potentially lead to more effective search.

## REPRODUCIBILITY STATEMENT

We release model checkpoints, training data, and code (including training and evaluation scripts) on our project website. Further training and evaluation details are provided in Appendix A.2 and A.3.

## ACKNOWLEDGMENTS

We thank Yu-Wei Chao, Lars Ankile, Alexander Swerdlow, Max Li, and the anonymous reviewers for their constructive comments which have helped improve various elements of this paper. We also thank Dan Blick, Sophia Huang, Mohammad Harrim, Yuzhu Dong, and Pooya Jannaty for their assistance with the OSS release of this project. This work was completed during an internship at NVIDIA in collaboration with Stanford University, and the first author was partially supported by the Robotics and AI Institute.

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

## A APPENDIX

### A.1 LATENT INJECTION IMPLEMENTATION DETAILS

As discussed in Section 4.1, Cosmos Policy learns to condition on and generate new non-image modalities (such as actions, robot proprioception, and values) through a mechanism called *latent injection*. Figure 8 provides a detailed visualization of how latent injection operates within Cosmos Policy's latent diffusion sequence.

The process begins with a sequence of images from multiple camera viewpoints, along with blank (all-zero) images that serve as placeholders for the new modalities to be injected. Once the video model's VAE tokenizer converts this image sequence into latent frames, we perform latent injection by overwriting the latent frames corresponding to the blank placeholder images with normalized and duplicated copies of the robot proprioception, action chunk, and value. Normalization rescales each modality to the range $[-1, +1]$, while duplication resolves the shape mismatch between the low-dimensional vectors and the target latent volumes.

For instance, consider an action chunk with shape $K \times d_{act}$. We first normalize each action dimension to $[-1, +1]$, and then flatten the array into a $(K \times d_{act})$ vector. This vector is duplicated $\frac{(H' \times W' \times C')}{(K \times d_{act})}$ times (where $(H', W', C')$ represents the shape of a single latent frame), reshaped into a $(H' \times W' \times C')$ volume, and used to overwrite the corresponding target latent frame. We apply an analogous process for robot proprioception and value, though their initial shapes differ.

Following latent injection, we corrupt portions of the latent sequence by adding randomly sampled Gaussian noise scaled by a randomly sampled noise level (see Appendix A.2.1 for details). Cosmos Policy is then trained to denoise these corrupted portions of the latent diffusion sequence while conditioning on the clean (uncorrupted) portions. Although latent injection introduces new modalities that were not present during the base video model's pretraining, we find that the model can still learn to generate them effectively through fine-tuning.

At inference time, Cosmos Policy generates clean (denoised) latent frames. Extracting the new modalities involves reversing the latent injection process described above. For example, to extract the action chunk prediction from the generated action latent, we compute the average action chunk across all $\frac{(H' \times W' \times C')}{(K \times d_{act})}$ copies in the latent volume. We then un-normalize the action chunk back to its original scale and deploy either the full action chunk or a portion of it on the robot. On the other hand, extracting the value is much simpler since it is a scalar (represented by a single floating point number): we simply take the average across the full latent volume and then un-normalize to the original range ($[0, +1]$). Note that extracting non-image modalities like these does not require any VAE decoding since these elements were directly injected into the latent space during training.

### A.2 TRAINING DETAILS

### A.2.1 COSMOS POLICY NOISE DISTRIBUTION

**Changes to noise levels during training and inference.** We find that the base Cosmos-Predict2 model's $\sigma$ (noise level) sampling scheme is suitable for video generation tasks but not the most effective for robot policy training. The latter requires generations to be very precise since the generated actions are used to directly control a robot and small imprecisions can lead to catastrophic failures. Therefore, to improve the accuracy of the generations at test time, we modify the noise level sampling scheme at both train and test time.

The base model's original noise distribution is a log-normal distribution, similar to the EDM formulation (Karras et al., 2022): $\ln(\sigma) \sim \mathcal{N}(P_{mean}, P_{std}^2)$ where $P_{mean} = 1.39, P_{std} = 1.2$ for Cosmos-Predict2-2B (see Figure 9, left). For action generation, we observe that the low weight on higher noise levels causes inaccurate action predictions during sampling. Diffusion generation begins with pure noise scaled by $\sigma_{max} = 80$ and iteratively denoises over multiple steps as $\sigma$ decreases to $\sigma_{min} \approx 0$. At each step, the network predicts the noise at the current $\sigma$ level to progressively recover a clean sample. Since the log-normal distribution concentrates training weight at lower noise levels, the model has insufficient signal at the high-$\sigma$ regime where generation begins, causing poor initial denoising and cascading errors. While this may not be critical for image and video generation, we find it is harmful for action generation, where predictions must be precise.

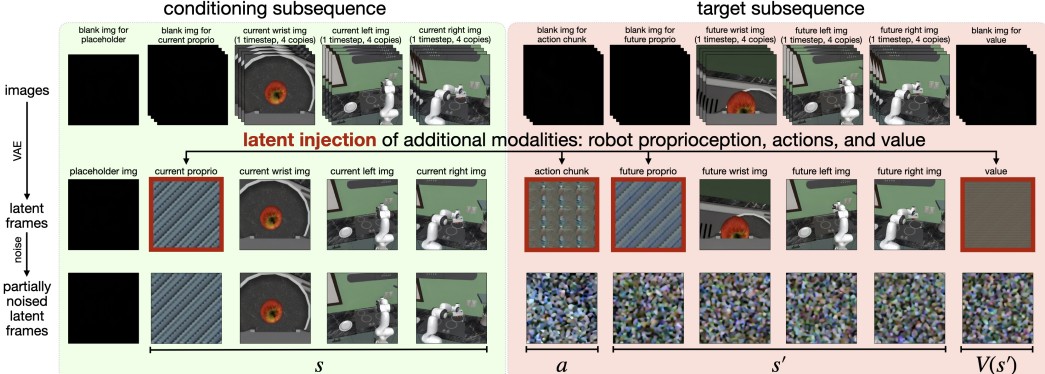

**Figure 8: Detailed view of the latent diffusion sequence of Cosmos Policy.** This is a more detailed version of Figure 2 which shows implementation details. As illustrated in the top row, blank (all zero) images are inserted into the input image sequence, and these images are encoded into latent frames by the VAE tokenizer. Then, in the middle row, these placeholder latent frames are completely overwritten by latent injection of new modalities (robot proprioception, actions, and value). Note that a placeholder image is set apart at the beginning of the sequence due to the video model's tokenization scheme, which encodes the first image by itself while the rest of the images are temporally compressed in groups of four (as discussed in Section 3). Therefore, to ensure that current timestep observations and future timestep observations have similarly structured latent representations, we place them after the blank first latent frame in the video diffusion sequence. In addition, since we wish to have one latent frame for each modality and camera viewpoint, we construct four identical copies of each image, as shown in the top row. Each group of four identical images corresponds to a single timestep rather than four timesteps).

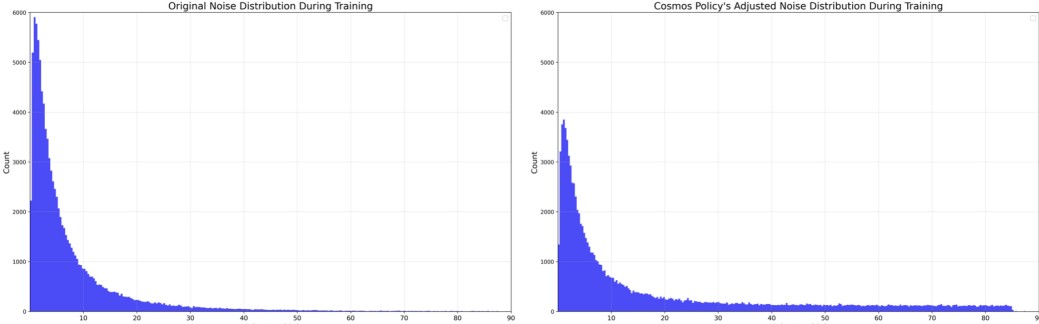

**Figure 9: Base model noise distribution vs. Cosmos Policy's adjusted noise distribution.** We change the base Cosmos-Predict2 video model's log-normal noise distribution (left) into a hybrid log-normal-uniform distribution (right) with more weight on higher noise levels. This modification empirically leads to more accurate generations at test time. See a detailed explanation in Appendix A.2.1.

Therefore, for Cosmos Policy training, we use a hybrid log-normal-uniform distribution with greater weight on larger noise levels (see Figure 9, right). To implement this, we sample from the original log-normal distribution with probability 0.7 and from a uniform distribution over $[1.0, 85.0]$ with probability 0.3, creating a log-normal distribution with an extended tail at higher $\sigma$ values. We find this empirically improves action prediction accuracy and overall success rate. We chose the 0.7/0.3 split to stay close to the original distribution while extending the high-$\sigma$ tail; these probability values were not tuned.

At test time, we find that the final denoising steps with $\sigma \approx 0$ are less accurate than earlier steps at larger $\sigma$ values, likely due to the low signal-to-noise ratio at very small noise levels. Therefore, when sampling from Cosmos Policy, we set a higher lower bound with $\sigma_{\min} = 4$ (rather than $\sigma_{\min} = 0.002$ as in the original EDM formulation (Karras et al., 2022)) while keeping $\sigma_{\max} = 80$. This higher lower bound empirically improves prediction accuracy at inference time for actions, future states, and values, as measured by lower L1 loss on training and validation samples.

### A.2.2 LIBERO TRAINING DETAILS

We train Cosmos Policy for 40K gradient steps using 64 H100 GPUs with global batch size 1920 (taking 48 hours total). The base Cosmos-Predict2 model is fully fine-tuned (all model weights). We train the policy to predict an action chunk size of 16 timesteps and execute the full chunk at test time before requerying. After 40K gradient steps, the policy's action L1 training loss is 0.012, future proprio L1 training loss is 0.007, future wrist image latent L1 training loss is 0.068, future third-person image latent L1 training loss is 0.036, and value function L1 loss is 0.007. All non-image modalities are normalized to the range $[-1, +1]$ for training. (Note that these are single-step training losses given varying $\sigma$ (noise levels) as input, rather than losses on generations from the multi-step diffusion sampling used during policy inference.) The magnitudes of these losses imply that the model more quickly learns to predict actions, proprioception, and values than images.

### A.2.3 ROBOCASA TRAINING DETAILS

We train Cosmos Policy for 45K gradient steps using 32 H100 GPUs with global batch size 800 (taking 48 hours total). The base Cosmos-Predict2 model is fully fine-tuned (all model weights). We train the policy to predict an action chunk of 32 timesteps and execute 16 steps before requerying. After 45K gradient steps, the policy's action L1 training loss is 0.016, future proprio L1 training loss is 0.007, future wrist image latent L1 training loss is 0.084, future third-person images latents L1 training loss is 0.048, and value function L1 loss is 0.007. All non-image modalities are normalized to the range $[-1, +1]$ for training.

### A.2.4 ALOHA TRAINING DETAILS

For Cosmos Policy and all other methods in our ALOHA robot evaluations, we train a single policy on all four tasks combined (185 total demonstrations). Cosmos Policy is trained for 50K gradient steps using 8 H100 GPUs with a global batch size of 200, taking 48 hours total. The base Cosmos-Predict2 model is fully fine-tuned (all model weights). We train with an action chunk size of 50 timesteps (spanning 2 seconds given a 25 Hz controller) and execute the full chunk before requerying. After 50K gradient steps, the policy's action L1 training loss is 0.010, future proprio L1 training loss is 0.008, future wrist images latents L1 training loss is 0.097, future third-person image latent L1 training loss is 0.085, and value function L1 loss is 0.007. All non-image modalities are normalized to the range $[-1, +1]$ for training.

For fair comparison with Cosmos Policy, we fine-tune $\pi_{0.5}$, $\pi_0$, and OpenVLA-OFT+ using the same computational budget: 48 hours on 8 H100 GPUs. Despite identical wall-clock time for fine-tuning, differences in training iteration speed across models and training frameworks result in substantial variation in the number of gradient steps completed: $\pi_{0.5}$ and $\pi_0$ are fine-tuned for 400K gradient steps (batch size 256), while OpenVLA-OFT+ is fine-tuned for 32K gradient steps (batch size 96 given gradient accumulation 4). We train for this seemingly large number of gradient steps across all methods because we observe that training loss continues to decrease and task execution quality generally improves with extended training. Finally, for the significantly smaller Diffusion Policy, which only contains approximately 150M parameters (as opposed to 2-7B for the other methods), we train from scratch on 1 H100 GPU for 48 hours, resulting in 190 epochs, equivalent to 72K gradient steps (batch size 350). Like Cosmos Policy, all policies are trained to predict an action chunk of 50 timesteps (except Diffusion Policy, which predicts 48 steps since it requires a multiple of 4), and the full chunk is executed before the model is requeried at test time.

### A.3 EVALUATION DETAILS

### A.3.1 GENERAL COSMOS POLICY EVALUATION DETAILS

**Number of denoising steps and parallel vs. autoregressive generation.** When evaluating Cosmos Policy as a direct policy, we predict actions, future state, and future state value altogether in parallel. Since direct policy evaluations do not use planning, we simply discard the future state and value predictions (though these can be saved for visualizations of the policy's intent and estimates of task progress). We use 5 denoising steps for LIBERO and RoboCasa and 10 denoising steps for ALOHA. On the other hand, when evaluating Cosmos Policy with model-based planning, we autoregressively generate actions, then future state, and then future state value. In the real-world evaluations with planning, we use 10 denoising steps for the actions, 5 for the future state, and 5 for the value. In

addition, we use an ensemble of 3 future state predictions and an ensemble of 5 value predictions to robustify the estimates.

### A.3.2 REAL-WORLD ALOHA ROBOT EVALUATION DETAILS

**Number of trials and initial conditions.** We evaluate every method in the real ALOHA robot experiments with 101 total trials: 30 trials for "put X on plate", 20 trials for "fold shirt", 25 trials for "put candies in bowl", and 26 trials for "put candy in ziploc bag". For each task, we test on both in-distribution and out-of-distribution (OOD) generalization scenarios with respect to the training demonstration dataset (see Figures 10 and 11 for visualizations). We describe the details of the test conditions below:

- "put X on plate": 20 in-distribution + 10 OOD trials. For in-distribution trials, the two objects (purple eggplant and brown chicken wing) are placed in the same row along the horizontal axis of the table, with varying position along the vertical axis. For OOD trials, the objects are not constrained to be in the same row and the positions are more varied. Max time limit is 350 timesteps (14 seconds).

- "fold shirt": 12 in-distribution + 8 OOD trials. For in-distribution trials, we use three T-shirts of different colors (blue with white stripes, plain white, and brown camouflage) with four fixed initial positions on the table (back, front, left middle, right middle). These three T-shirts were seen at data collection time (five demos for each of the three shirt, for 15 training demos total). For OOD trials, we use an unseen pink T-shirt with the same four initial positions for the first four trials, and the seen blue-with-white-stripes shirt with unseen distractor objects placed on the table (other shirts folded up and placed on the sides or corners of the table, serving as visual distractors). Max time limit is 1600 timesteps (64 seconds).

- "put candies in bowl": 15 in-distribution + 10 OOD trials. For in-distribution trials, we place 5 candies in front of a large metal bowl, with a roughly even balance of candies on the left versus right half of the table. For OOD trials, we either (a) place 5 candies in front or behind a large metal bowl with uneven amounts of candies on the left versus right half, or (b) place 5 candies in front or behind an unseen smaller orange bowl. Max time limit is 1100 timesteps (44 seconds).

- "put candy in ziploc bag": 20 in-distribution trials + 6 OOD trials. For in-distribution trials, we place a pink ziploc bag filled half full with candies in the middle of the table with varying initial position along the vertical axis, and one piece of candy to its right (also varying along the vertical axis). For OOD trials, we replace the pink ziploc bag with an unseen blue ziploc bag that is filled to about 75 percent full (more than seen during training), which requires more firm grasps when grasping the bag so that it does not slip out of grasp as the robot manipulates it. Max time limit is 1100 timesteps (44 seconds).

**Scoring rubrics.** Here we describe how each policy rollout is scored between 0 and 100 points. Each score represents what percent of the task has been successfully completed. We use score instead of success rate since a binary metric does not capture fine-grained details.

- "put X on plate": 50 points for touching the correct target object. 50 points for putting the correct target object on the plate.

- "fold shirt": 10 points for each of the following stages: (1) grasped bottom edge of shirt, (2) folded in half, (3) grasped both sleeves, (4) folded sleeves in, (5) grasped bottom edge of shirt, (6) folded in half again, (7) grasped right edge of shirt, (8) folded in half again, (9) released the shirt from grasp, (10) pushed shirt to center of table. In the end, 10 points are deducted if a part of the shirt (e.g. sleeve) sticks out due to an incomplete fold.

- "put candies in bowl": 20 points for each of 5 candies grasped and placed into the bowl.

- "put candy in ziploc bag": 20 points for each of the following stages: (1) grasped ziploc bag opener with right arm, (2) grasped ziploc bag's upper-left corner with left arm, (3) opened bag at least halfway by pulling the opener with right arm, (4) grasped candy with right arm, (5) placed candy inside bag.

## Initial Positions for In-Distribution Trials

"put X on plate"

"fold shirt"

"put candies in bowl"

"put candy in ziploc bag"

**Figure 10: In-distribution initial conditions for ALOHA robot evaluations.** Here we show sample initial positions used for evaluating policies in conditions similar to the training demonstrations.

**Detailed in-distribution vs. OOD evaluation scores.** Table 3 shows the breakdown of in-distribution and out-of-distribution scores for every method across the ALOHA robot task suite. Cosmos Policy achieves highest aggregate success rates, though $\pi_{0.5}$ shows slightly better performance specifically in OOD test scenarios.

**Table 3: In-distribution vs. OOD evaluation scores for ALOHA robot evaluations.** This table is a a more detailed summary of the results shown in Figure 4.

| Method | "put X on plate" Score | "fold shirt" Score | "put candies in bowl" Score | "put candy in ziploc bag" Score | Average Score |
|---|---|---|---|---|---|
| **In-distribution** | | | | | |
| Diffusion Policy | 85.0 | 23.3 | 37.3 | 11.0 | 39.2 |
| OpenVLA-OFT+ | 87.5 | 99.2 | 22.7 | 59.0 | 67.1 |
| $\pi_0$ | 97.5 | 98.3 | 73.3 | 56.0 | 81.3 |
| $\pi_{0.5}$ | 97.5 | 99.2 | 98.7 | 56.0 | 87.8 |
| Cosmos Policy | 100.0 | 99.2 | 100.0 | 86.0 | 96.3 |
| **OOD** | | | | | |
| Diffusion Policy | 20.0 | 23.8 | 26.0 | 26.7 | 24.1 |
| OpenVLA-OFT+ | 30.0 | 100.0 | 20.0 | 56.7 | 51.7 |
| $\pi_0$ | 60.0 | 98.8 | 68.0 | 60.0 | 71.7 |
| $\pi_{0.5}$ | 100.0 | 100.0 | 90.0 | 80.0 | 92.5 |
| Cosmos Policy | 100.0 | 100.0 | 74.0 | 83.3 | 89.3 |
| **Full (In-distribution + OOD)** | | | | | |
| Diffusion Policy | 63.3 | 23.5 | 32.8 | 14.6 | 33.6 |
| OpenVLA-OFT+ | 68.3 | 99.5 | 21.6 | 58.5 | 62.0 |
| $\pi_0$ | 85.0 | 98.5 | 71.2 | 56.9 | 77.9 |
| $\pi_{0.5}$ | 98.3 | 99.5 | 95.2 | 61.5 | 88.6 |
| Cosmos Policy | 100.0 | 99.5 | 89.6 | 85.4 | 93.6 |

## Initial Positions for Out-of-Distribution Trials

**Figure 11: Out-of-distribution initial conditions for ALOHA robot evaluations.** Here we show sample initial positions used for evaluating policies in unseen test conditions.

**Table 4: Cosmos Policy ablations in LIBERO.** Here we report the results of two independent ablations: (1) In Section 4.2, we discussed that Cosmos Policy's policy and world model training involves additional targets which provide auxiliary supervision: the policy learns to jointly predict $p(a, s', v(s'))|s)$ instead of $p(a|s)$, and the world model learns to jointly predict $p(s', V(s')|s, a)$ instead of $p(s'|s, a)$. To evaluate the effect of this joint learning objective, we train a version of Cosmos Policy without it by masking the loss on the additional targets: the policy predicts only $p(a|s)$ and the world model predicts only $p(s'|s, a)$ (the value function continues to predict $p(V(s')|s, a, s')$ as before). We find that removing the auxiliary objectives leads to a 1.5 point drop in average success rate. (2) We also train a version of Cosmos Policy from scratch and observe a 3.9 point drop from the full version.

|  | Spatial SR (%) | Object SR (%) | Goal SR (%) | Long SR (%) | Average SR (%) |
|---|---|---|---|---|---|
| Cosmos Policy (ours) | 98.1 | 100.0 | 98.2 | 97.6 | 98.5 |
| (1) w/o auxiliary losses | 97.6 | 99.8 | 96.7 | 94.0 | 97.0 |
| (2) w/o pretrained model | 94.7 | 98.9 | 96.3 | 88.6 | 94.6 |

## A.4 ADDITIONAL EXPERIMENTS AND DETAILS

### A.4.1 ADDITIONAL ABLATION EXPERIMENTS

To further study the effects of individual components of the Cosmos Policy design, as well as the joint training objectives discussed in Section 4.2, we conduct a series of additional ablation experiments that remove individual components or objectives one at a time, until we ultimately have a barebones policy that only predicts actions (no future state or value). Recall from Section 4.2 that Cosmos Policy is trained with a joint objective that optimizes the policy, world model, and value function altogether in one architecture. Specifically, each batch of training samples is split 50/25/25: 50 percent of the samples are used for training the policy, 25 percent for the world model, and 25 percent for the value function. This balanced splitting determines which parts of the latent diffusion are used as conditioning and which parts are used as the target, which in turn determines whether we are training the policy, the world model, or the value function (see Figure 12 for an illustration). In addition to these balanced batches, the policy and world model are trained with auxiliary objectives: the policy learns to jointly predict $p(a, s', v(s'))|s)$ instead of $p(a|s)$, and the world model learns to jointly predict $p(s', V(s')|s, a)$ instead of $p(s'|s, a)$

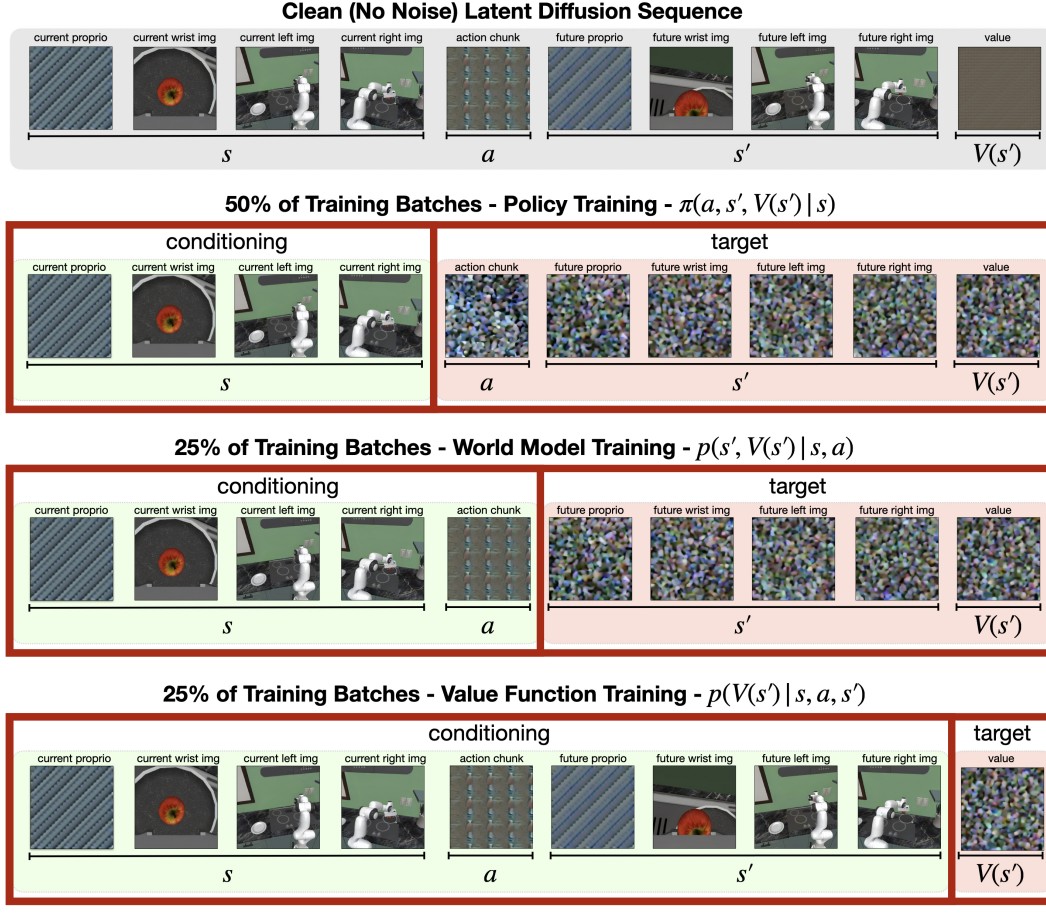

**Figure 12: Cosmos Policy balanced batches training scheme.** We illustrate the joint objectives training scheme discussed in Section 4.2. While each batch of training samples is split 50/25/25 for policy, world model, and value function training, respectively, the full latent diffusion sequence remains fixed, and the conditioning scheme determines which of the three functions is being optimized. During training, the model is tasked to denoise the target noised latent frames conditioned on the clean latent frames. Note that the above depicts the initial base policy training scheme. When optionally refining the world model and value function on policy rollouts (in preparation for model-based planning), we mask out the current state and action during value function training so that the value predictions are only a function of the future state $s'$. Separately, to train a Q-value function, we can instead mask out the future state so that the value predictions are only a function of the current state and action, $(s, a)$; this variant can be used for model-free planning (though we find that the model-based planning variant performs better, as shown in Figure 7).

### A.4.2 COSMOS POLICY INFERENCE LATENCY

Here we discuss measurements of inference latency in terms of wall-clock time. Cosmos Policy generates action, future state, and value predictions in parallel in 0.61 seconds total on 1 H100 GPU when sampling with 5 denoising steps (for LIBERO and RoboCasa). Inference latency increases to 0.95 seconds on 1 H100 GPU when sampling with 10 denoising steps (for ALOHA). Inference latency decreases to 0.16 seconds on 1 H100 GPU when sampling with 1 denoising step. We observe competitive performance in RoboCasa (66.4% average success rate) even with 1 denoising step, as shown in Table 5. Note that these latencies are the time spent generating an action *chunk*. For example, on the ALOHA robot, the robot pauses for 0.95 seconds to generate an action chunk that spans 2 seconds of execution.

For model-based planning experiments on the real ALOHA robot, we do best-of-N search with $N = 8$ using 8 parallel H100 GPUs (1 for each branch in the search). Cosmos Policy first generates $N$ candidate action chunks with 10 denoising steps each, then generates an ensemble of 3 future state predictions per action proposal with 5 denoising steps each, and then generates an ensemble of

**Table 5: Cosmos Policy ablations and additional experiments in RoboCasa. Top:** We ablate individual components of the joint objectives training scheme and auxiliary supervision discussed in Section 4.2 and visualized in Figure 12. Each version of the policy is trained with the exact same training hyperparameters and compute as the original Cosmos Policy in RoboCasa, and evaluated across all 24 tasks with 50 trials per task and 3 random seeds (3600 trials total). We remove one component at a time until we arrive at a barebones policy that only predicts actions. Here are details describing each ablation: (1) We ablate value function training samples (last row of Figure 12) and instead use 50% of batches for policy training and 50% for world model training. (2) We ablate both world model and value function training samples (last two rows of Figure 12) and instead use 100% of each training batch for policy training only. (3) Adding to the prior ablation, we also ablate the value target when training the policy, thus optimizing the policy to learn just $\pi(a, s'|s)$ rather than $\pi(a, s', V(s')|s)$. (4) Adding to the prior ablation, we ablate both the future state and value targets when training the policy, thus training a barebones policy that only models $\pi(a|s)$. We observe the most significant drop in performance from the final ablation, suggesting that training the policy to also predict future state is crucial to the effectiveness of Cosmos Policy. **Bottom:** We also report the performance of Cosmos Policy with just 1 denoising step per action chunk at inference time. We observe a 66.4% average success rate with just 0.16 second inference latency per action chunk of 32 timesteps on one H100 GPU, which is nearly 4× faster than the 0.61 second latency observed with 5 denoising steps with only a small hit (-0.5%) on success rate.

| | Average SR (%) |
|---|---|
| Cosmos Policy (ours) (5 denoising steps) | 67.1 |
| (1) w/o value function training samples | 66.6 |
| (2) w/o world model and value function training samples | 64.0 |
| (3) w/o world model and value function training samples & auxiliary value supervision for policy training samples | 62.5 |
| (4) w/o world model and value function training samples & auxiliary future state and value supervision for policy training samples | 44.4 |
| Cosmos Policy w/ 1 denoising step | 66.4 |

5 value predictions per future state prediction with 5 denoising steps each. In total, this search takes 4.9 seconds on 8 parallel H100 GPUs.

Though the high inference latency did not prevent effective task execution in our evaluations, we acknowledge the limitations caused by slow inference, such as difficulty in adapting the method to dynamic manipulation tasks or locomotion tasks. Optimizing the inference speed for the policy and the model-based search process would be a promising avenue for future work in order to enable adoption to a wide variety of robotics applications.

