# OpenReview forum: "Cosmos Policy: Fine-Tuning Video Models for Visuomotor Control and Planning"
_ICLR.cc/2026/Conference — ICLR 2026 Poster_

### Official Review · Reviewer_pF5s · 2025-11-01

**Soundness:** 3
**Presentation:** 2
**Contribution:** 2
**Rating:** 4
**Confidence:** 4

**Summary:**

This paper proposes a Cosmos policy framework that is built upon the Cosmos pre-trained video model. To add required modalities to the pre-trained model, this paper constructs additional latent frames containing the information. Experiments are conducted on the Libero tasks and the real Aloha task, where the proposed policy obtains better performance on average.

**Strengths:**

- This paper proposes fine-tuning a pre-trained video model to a policy that can execute and interact with the real environment. The idea and direction are very promising and interesting for the community.
- Instead of adding an additional network for integrating modalities like actions, the paper constructs latent frames to better leverage the prior learned in the pre-trained model. While this idea is not novel, it is valuable for robotic policy learning.
- Experiments include both simulation environments and real tasks.

**Weaknesses:**

- While I like the research topic studied in this paper, the proposed method is not novel and not elegant enough. For example, an executable policy requires high frequency, which makes small policy networks and action chunking useful. However, this paper requires the Cosmos pre-trained model to output executable, low-level actions directly, which can cause high latency and make it difficult to adapt to other tasks like locomotion. The second point is that the latent frames contain duplicate information, which can cause learning inefficiency. It is not necessary for the robot's proprio states and actions to occupy several latent frames. Overall, the proposed fine-tuning pipeline is not easy to reproduce, and still needs to be improved.
- While the proposed policy gets the best success rates on average, its performance on o.o.d. tasks is not demonstrated clearly. Can the policy generalize to different tasks with the same action space? Can the policy generalize to different backgrounds and unseen objects? Since the policy is built upon a pre-trained model, I expect it to generalize better.
- Missing citations related to fine-tuning the video prediction model to an executable policy, e.g., Learning an Actionable Discrete Diffusion Policy via Large-Scale Actionless Video Pre-Training, NeurIPS 2024.
- Missing Reproducibility statement.

**Questions:**

- Can the proposed framework also work well for other pre-trained video models, especially considering different architectures?
- Please address my concerns in Weaknesses.

---

> ### Author Response · Authors · 2025-11-21
> **Response to Reviewer pF5s**
>
> We thank reviewer **pF5s** for the constructive comments, which have helped us to improve the paper through new experiments. We address specific comments below.
>
> > While the proposed policy gets the best success rates on average, its performance on o.o.d. tasks is not demonstrated clearly. Can the policy generalize to different tasks with the same action space? Can the policy generalize to different backgrounds and unseen objects?
>
> Please see our General Response for a **new experiment that evaluates Cosmos Policy on unseen objects and scenes**. We find that Cosmos Policy can indeed generalize to these conditions and actually achieves state-of-the-art performance with much fewer demonstrations than prior approaches. In addition, we have clarified the evaluations in our prior ALOHA robot experiments by adding significantly more detail in Appendix A.3.2. Our ALOHA robot evaluations already involve out-of-distribution generalization tests, such as manipulating unseen objects and handling unseen object arrangements. Our results demonstrate that Cosmos Policy can indeed generalize to unseen test conditions in both simulated and real-world environments.
>
> > The second point is that the latent frames contain duplicate information, which can cause learning inefficiency.
>
> While it is true that newly injected latent frames contain duplicate information, we find that this empirically does not cause learning inefficiency. In fact, we use the **same compute budget** (48 hours on 8 H100 GPUs) to train Cosmos Policy and prior methods such as pi0.5 and OpenVLA-OFT+ on the real-world ALOHA demonstrations, and we find that Cosmos Policy achieves superior overall performance. We add this clarification regarding equal compute budget to Appendix A.2.4. In addition, our new RoboCasa simulation experiment described in the General Response further demonstrates superior learning efficiency, as Cosmos Policy requires **significantly fewer training demonstrations** than prior works to achieve similar success rates.
>
> > While I like the research topic studied in this paper, the proposed method is not novel and not elegant enough.
>
> We believe that our proposed method is **novel** because it is the first to post-train a large-scale pretrained video foundation model into a unified policy, world model, and value function that can jointly generate robot actions, future state images, and values, respectively — with zero architectural modifications. In contrast, prior video-based policies either require designing new action modules (e.g. Video Prediction Policy and Video Policy) or train from scratch without video priors due to incompatibility in design between pretrained video model and the target unified architecture (e.g. UVA and UWM). We also believe that it is **elegant** due to its simplicity: no architectural modifications; one unified architecture for policy, world model, and value function; and a single stage of post-training (versus prior works which use two training stages: one for fine-tuning the video model on target domain-specific robot data and another for training the action module). The simplicity in design is not merely an aesthetic but is rather accompanied by strong performance, with Cosmos Policy achieving state-of-the-art success rates in multiple domains.
>
> > Missing Reproducibility statement.
>
> We have added a Reproducibility Statement to the paper. Thank you for mentioning this.
>
> > Missing citations related to fine-tuning the video prediction model to an executable policy, e.g., Learning an Actionable Discrete Diffusion Policy via Large-Scale Actionless Video Pre-Training, NeurIPS 2024.
>
> Thank you for letting us know. We have added this citation to the Related Work section.
>
> > Can the proposed framework also work well for other pre-trained video models, especially considering different architectures?
>
> Although we have not tried post-training other pretrained video models in this work, several video models use a similar latent diffusion scheme and VAE tokenization scheme as Cosmos-Predict2, which suggests applicability to other pretrained models. For example, both CogVideoX and Wan2.1 use the same VAE compression scheme with 4x temporal compression and 8x spatial compression and perform diffusion in the image latent space. Since our latent injection scheme is agnostic of the underlying architecture (we merely inject additional frames into the diffusion latent space), it is compatible with other base video models.

---

> > ### Comment · Reviewer_pF5s · 2025-11-27
> >
> > Thanks for the authors' response. The added experiments and detailed clarifications have addressed my concerns regarding the generalization and learning efficiency. However, I still remain concerned about the high inference latency, which hinders its real-world applicability. Additionally, I am still not sure whether the method can work well for other video generative models like Wan, since these pre-trained models are not explicitly trained with a large-scale robot dataset.

---

> ### Author Response · Authors · 2025-12-03
> **Follow-Up Response to Reviewer pF5s**
>
> Thank you to reviewer **pF5s** for confirming that our new experiments and clarifications have addressed concerns about generalization and learning efficiency.
>
> We address the two remaining concerns below:
>
> > I still remain concerned about the high inference latency
>
> We add a **follow-up experiment** in which we evaluate Cosmos Policy in the 24 RoboCasa tasks with **just 1 denoising step** (instead of the default 5-10 steps) to speed up inference, using the same checkpoint from our General Response. The flexibility in number of denoising steps at inference time is possible due to the continuous noise formulation of the Cosmos-Predict2 base model, which follows the EDM diffusion modeling framework [1]. We observe a **66.4% average success rate** with just **0.16 second inference latency** per action chunk of H=32 timesteps on 1 H100 GPU (note: changing H does not affect latency since we generate a full chunk in one latent frame regardless of the chunk size). For reference, our earlier result showed 67.1% average success rate with 0.61 second latency using 5 denoising steps; in comparison, our new result with 1 denoising step shows nearly 4x faster inference with only a 0.5% drop in success rate. We add these results to Table 5 in the Appendix.
>
> If we use this setting on the real ALOHA robot seen in our prior experiments, we would observe a small 0.16 second pause between action chunks that span 2 seconds of execution each, which is acceptable for most real-world manipulation tasks in practice. Lastly, we note that many optimizations can be performed to increase inference speed, such as model compilation, quantization, distillation, and sparse attention, none of which we explored in this work since they fall outside the focus of this paper.
>
> > I am still not sure whether the method can work well for other video generative models like Wan, since these pre-trained models are not explicitly trained with a large-scale robot dataset.
>
> While using a base video model such as Cosmos-Predict2 that is pretrained on large-scale robot video data is beneficial, this is not a hard requirement for our approach. Empirical evidence for this is shown in our ablation study in Table 4, which reveals that **even when training Cosmos Policy from scratch** (with randomly initialized weights instead of the pretrained Cosmos-Predict2 video model weights), our method **still achieves competitive performance**: 94.6% average success rate in LIBERO, which surpasses the success rates of strong prior methods such as the fine-tuned $\pi_0$ VLA (94.2%), Dita (92.3%), and Diffusion Policy (72.4%), as shown in Table 1. Note that this from-scratch version is a lower bound on the performance: other base video models, even if they do not benefit from large-scale robot data pretraining, would still bring strong spatial and temporal modeling priors from large-scale Internet video data and enable the policy to far outperform random initialization. We leave empirical validation of this hypothesis to future work.
>
> We appreciate your engagement with our work and hope that these additional experiments address your remaining concerns.
>
> [1] Karras et al. Elucidating the design space of diffusion-based generative models. NeurIPS 2022.

---

### Official Review · Reviewer_nGa7 · 2025-11-02

**Soundness:** 3
**Presentation:** 3
**Contribution:** 3
**Rating:** 6
**Confidence:** 4

**Summary:**

The authors propose a way to use a video generation model for robot control through finetuning the model without multi-stage post-training or architectural modifications to the model.  Future latent frames are decoded into future images and cumulative reward values.  This enables test-time planning of action trajectories. Simulation results are presented on the LIBERO sim benchmark.

**Strengths:**

- The main strength of this work is in using the video generation model to output actions with only a single fine-tuning stage without any architectural changes to the model - this is in contrast to other approaches that employ architectural changes like adding inverse dynamics model or doing multiple stages of post-training to generate actions.

- The proposed approach enables model-based planning where multiple action proposals can be sampled from the policy and resulting states / values can be predicted for each action sequence and the action sequence with the highest value can be selected.

**Weaknesses:**

- It's unclear why making architectural changes to the video generation model is seen as a weakness in other approaches.

- Are the authors assuming the world model is on the state s (in Sec. 3)? They authors claim the world model predicts the state.  World models predict observations and not states.  Clearly making this distinction is important as the state is not completely observable. The method in the paper predicts value function as a function of the state.  However, since we are only predicting the observation, the value function can't be a function of this observation but rather a function of the (unknown) state.  Does this break things in the formulation?

**Questions:**

- The authors claim that they can predict videos for new camera views as well as states and value functions from the same model without architectural changes.  This is done through latent frame injection.  Sec. 4.1 does not provide sufficient detail on how this is done.  In particular, what should the latent state injection be.  Blank or copies of current latents should not work.  Can the authors add more details on how this is enabled - is the finetuning on the entire network rather than just a few layers?  Would the latent injection not cause an increase to the size of the latent input and an increase to the network size for the subsequent layers?  If so, is this not considered an architectural change?

- Since the work depends significantly on the pre-trained video model, Cosmos-Predict2, it might be good to provide an overview and key features and capabilities of this model.  This can be added as an appendix.  Particularly, it will be interesting to point out the differences between this video model and other video models - this will enable readers to see if they can start with their own existing video models.

---

> ### Author Response · Authors · 2025-11-21
> **Response to Reviewer nGa7**
>
> We thank reviewer **nGa7** for the thoughtful and constructive feedback, which we have used to improve the clarity of several sections in our paper. Please find our responses to specific concerns below.
>
> > The authors claim that they can predict videos for new camera views as well as states and value functions from the same model without architectural changes. This is done through latent frame injection. Sec. 4.1 does not provide sufficient detail on how this is done. In particular, what should the latent state injection be.
>
> Thank you for the helpful feedback. We have added significantly greater detail regarding the latent injection approach to Appendix A.1, with extended explanations and implementation details. We hope that this improves clarity, though please do not hesitate to ask further questions if anything remains unclear.
>
> > It's unclear why making architectural changes to the video generation model is seen as a weakness in other approaches.
>
> While making architectural changes is not inherently a weakness of other approaches, it introduces some practical complexities that our method avoids entirely to achieve both simplicity and strong policy performance. For example, consider creating a separate action diffuser network or inverse dynamics model to generate actions, as done in prior works. This requires one to (1) design the new architecture; (2) decide on the algorithm for predicting actions (e.g., token generation, regression, diffusion, etc.); and (3) coordinate the training of the video model and the action module (in either separate training stages, which requires deciding how long each stage should be, or one joint learning stage, which requires tuning the final objective to avoid learning instability). All of this would likely be accompanied by careful hyperparameter tuning to ensure that the new design is effective and its optimization is stable. On the other hand, our method avoids these complexities by inheriting the pretrained video model's high-capacity architecture and expressing learning algorithm to learn actions and other modalities. This is simpler to implement and automatically benefits from the video model's proven ability to capture complex distributions, as shown in our state-of-the-art results.
>
> > Are the authors assuming the world model is on the state s (in Sec. 3)? They authors claim the world model predicts the state. World models predict observations and not states. Clearly making this distinction is important as the state is not completely observable.
>
> Thank you for pointing this out. We have added a note to Section 3 clarifying that the world model technically predicts future observations, since the true state is not completely observable. To be precise, the value function is indeed a function of the predicted future observations (though we opt to use the term “state” to improve notational simplicity and readability throughout the paper).
>
> > is the finetuning on the entire network rather than just a few layers?
>
> Thank you for reminding us to include this detail. We fully fine-tune the entire network. We have added this note to Appendix A.2.

---

### Official Review · Reviewer_i7f5 · 2025-11-03

**Soundness:** 3
**Presentation:** 3
**Contribution:** 3
**Rating:** 8
**Confidence:** 4

**Summary:**

This paper introduces Cosmos Policy, which adapts a pretrained video generative model into a robot policy though post-training without architectural modifications. In order to adapt the pretrained video model (image+text -> video) to a robot policy (requires additional modalities including robot proprioception, robot actions, state values, multiple camera views....), the authors propose to encode the additional modalities as additional latents. This simple design enables the joint training of policy, world model and value function, which can be leveraged appropriately to perform the demanded robot task e.g., direct policy evaluation, or evaluation with planning. Experiments show strong performances compared to existing VLAs or robot policy models, across the single-arm LIBERO simulation and two-arm ALOHA platform.

**Strengths:**

- Strong empirical performances across the evaluation benchmarks, even existing methods which already rely on video generative models.

- The proposed idea is simple yet effective, and ebles joint training of policy, world model and value function within the same design.

- Sufficient analyses and ablation results (e.g., w/o auxiliary losses, Q_sa and V_s variants) which show interesting and significant results.

**Weaknesses:**

- I understand that the additional modalities are encoded as additional latents, but I still can't understand exactly how. I can understand from Figure 1 that the different modalities are interleaved, and the current state frames are given as conditioning inputs - but it is hard to understand where the original latents remain, and where the additional modalities are input.

- Without the pretrained model, the performance of Cosmos Policy falls below that of CogVLA. What happens if the same post-training scheme of Cosmos Policy is applied to CogVideo/CogVideoX models? This would be a fairer comparison, as the newer Cosmos-Predict2-2B-Video2World is generally considered to be a stronger model.

**Questions:**

- What happens if the same post-training scheme of Cosmos Policy is applied to CogVideo/CogVideoX models?

- How exactly are the additional modalities being encoded into latents are are being interleaved with the original latents?

---

> ### Public Comment · ~Jaemin_Cho1 · 2025-11-20
>
> Dear authors - nice work! It was a good read. I got the same question with this reviewer. How exactly are the additional modalities, especially action, being encoded into and decoded from latents? Do you plan to share public code? This is crucial detail for anyone who wants to reproduce your experiments, and I'd love to try this method.

---

> > ### Author Response · Authors · 2025-11-21
> > **Response to Public Comment**
> >
> > Thank you for reading our paper and leaving a positive comment!
> >
> > > How exactly are the additional modalities, especially action, being encoded into and decoded from latents?
> >
> > Please see the response to reviewer i7f5 that we recently posted. In particular, Appendix A.1 in our revised submission includes implementation details that should answer your first question regarding the injection and extraction of additional modalities (though please do not hesitate to request clarification if something is still unclear).
> >
> > > Do you plan to share public code?
> >
> > Yes, we will share training and evaluation code as well as model checkpoints upon publication so that researchers can reproduce our experiments.

---

> ### Author Response · Authors · 2025-11-21
> **Response to Reviewer i7f5**
>
> We thank reviewer **i7f5** for the thoughtful review and helpful comments. We respond to individual concerns below.
>
> > I understand that the additional modalities are encoded as additional latents, but I still can't understand exactly how. I can understand from Figure 1 that the different modalities are interleaved, and the current state frames are given as conditioning inputs - but it is hard to understand where the original latents remain, and where the additional modalities are input.
>
> To improve the clarity of Cosmos Policy’s latent injection mechanism, we have added significantly more details (including technical implementation details) about the process to Appendix A.1; we simplified the original latent diffusion sequence figure (Figure 2); and we added a new detailed version of the latent diffusion sequence figure (Figure 8). We hope that this helps with understanding, though please do not hesitate to ask further questions if anything is unclear.
>
> > How exactly are the additional modalities being encoded into latents are are being interleaved with the original latents?
>
> (See previous answer.)
>
> > What happens if the same post-training scheme of Cosmos Policy is applied to CogVideo/CogVideoX models?
>
> Although we have not tried post-training the CogVideoX video model, this model follows the same latent diffusion transformer structure as Cosmos-Predict2 and even uses the same VAE compression scheme with 4x temporal compression and 8x spatial compression (this is a commonly used tokenization scheme among recent video diffusion models, even including the Wan models). Due to these similarities and because our simple latent injection approach is agnostic to the base model, we expect the same post-training scheme to work on CogVideoX as well.

---

> > ### Comment · Reviewer_i7f5 · 2025-11-26
> >
> > I appreciate the authors' response to my comments.
> >
> > 1. Cosmos Policy's latent injection mechanism has become much clearer and more straightforward to understand.
> >
> > 2. Thank you for the response to my question regarding CogVideo/CogVideoX models. I agree with the authors, but I still believe it would be interesting to see how the proposed method works under different architectures / model sizes.
> >
> > I still believe this is an interesting and strong paper, and would like to maintain my current rating.

---

### Official Review · Reviewer_nsat · 2025-11-03

**Soundness:** 3
**Presentation:** 3
**Contribution:** 3
**Rating:** 8
**Confidence:** 4

**Summary:**

This paper presents Cosmos Policy, a method for adapting a large pretrained video diffusion model (Cosmos-Predict2) for "world modeling" in visuomotor policy learning. Specifically, they use a pretrained video diffusion model which takes in a sequence of latent frames to produce RGB frames as output to also take in new types of latent frames (corresponding to robot actions, states, and value function) and produce the corresponding future outputs. Instead of adding new networks or heads (which most prior works do), Cosmos Policy reuses the exact same video diffusion transformer and pretends that these new signals (actions, proprioception, values) are just additional “frames” in the video sequence. Cosmos Policy thus jointly learns to denoise actions, future states, and expected returns, leveraging the pretrained model’s spatiotemporal priors for physically grounded control.

The paper further demonstrates a model-based planning extension, where a fine-tuned world model and value function are used to evaluate multiple action proposals in a “best-of-N” search procedure. Experiments in simulation and one some real-world manipulation tasks shows the method outperforming both diffusion-based policies trained from scratch and fine-tuned vision-language action (VLA) models

**Strengths:**

I like the idea of repurposing a video generation model (that has already learned spatio-temporal predictions) for other spatio-temporal prediction tasks, in this case robot action/value data. The evidence provided by the paper that this simple idea works is noteworthy beyond just the numbers. The numbers themselves are impressive where fine-tuning the 1.2B-parameter Cosmos-Predict2 model on just a few hundred robot demonstrations yields 98.5% task success on the LIBERO benchmark, outperforming both diffusion-based and VLA baselines (e.g., Pi0, OpenVLA) trained from scratch.

I also like the extension to model-based planning without requiring architectural changes. I think this idea can be further exploited by predicting more spatio-temporal signals useful for model-based planning.

**Weaknesses:**

1. It would have been nice to see specific numbers on how well the state, actions, and value functions are predicted using this model.

2. It is also not clear how well this would generalize standard tasks (that may be in the pretraining of the underlying model.. even though I fully agree that the action data is not). But seeing more generalizability experiments would have been nice.

3. It was not clear how well does this work for longer-horizon tasks.

**Questions:**

1. Unless I missed it, I didn't see any ablations on other ways of encoding the state, action, value information as latent frames. For example, could you combine those in different ways? How much would the training/finetuning pipeline have to changes if we were to change the embodiment, for example?

2. I would like to see more discussion on the computational overhead. How much does this add during inference time and how realistic is that for control in real-time on hardware?

3. How well would this generalize for longer-horizon tasks?

4. Do you have any tools or visualizations to understand how the injected latent frames influence downstream denoising? For example, does the diffusion noise schedule or attention pattern shift when action/value latents are introduced?

---

> ### Author Response · Authors · 2025-11-21
> **Response to Reviewer nsat**
>
> We thank Reviewer **nsat** for recognizing both the simplicity and effectiveness of our approach in treating actions, states, and values as additional latent frames in a unified video diffusion model. We address specific concerns and questions below:
>
> > It is also not clear how well this would generalize standard tasks (that may be in the pretraining of the underlying model.. even though I fully agree that the action data is not). But seeing more generalizability experiments would have been nice.
>
> We have added new generalization experiments conducted in the RoboCasa simulation benchmark to the revised submission (please see the General Response above). In particular, we observe that Cosmos Policy generalizes to unseen object instances and environment scenes/styles and outperforms prior state-of-the-art methods with far fewer training demonstrations.
>
> > Unless I missed it, I didn't see any ablations on other ways of encoding the state, action, value information as latent frames. For example, could you combine those in different ways?
>
> Thank you for this helpful suggestion. We have added new ablation experiments that combine the actions, future state, and value in different ways; please see the General Response above.
>
> > It would have been nice to see specific numbers on how well the state, actions, and value functions are predicted using this model.
>
> Thank you for the suggestion. We have added these details to the Appendix in the revised submission (see Appendices A.2.2, A.2.3, and A.2.4). For example, for our recent RoboCasa experiments, we observe the following final training losses from Cosmos Policy: action L1 loss is 0.016, future proprio L1 loss is 0.007, future wrist image L1 loss is 0.084, future third-person images L1 loss is 0.048, and value function L1 is 0.007. The policy evidently fits the multimodal dataset quite effectively, and it seems to learn non-image modalities faster than the image modalities (as images are more difficult to predict).
>
> > Do you have any tools or visualizations to understand how the injected latent frames influence downstream denoising? For example, does the diffusion noise schedule or attention pattern shift when action/value latents are introduced?
>
> Yes, we have added implementation details regarding diffusion noise sampling and a new visualization (Figure 9) with discussion in Appendix A.2.1. In short, since action generation must be precise for effective low-level robotic control, we have adjusted the noise schedule during training so that higher noise levels are sampled. The new Figure 9 illustrates the modifications.

---

### Author Response · Authors · 2025-11-21
**General Response**

# General Response to Reviewers

We thank reviewers **nsat**, **i7f5**, **nGa7**, and **pF5s** for their constructive comments, which have helped us to improve the paper (revised draft now uploaded - updates addressing concerns highlighted in red).

## New Experiments and State-of-the-Art Results
We have added new comparisons highlighting Cosmos Policy’s effectiveness in additional settings:
* **Generalization experiments in RoboCasa simulation benchmark [pF5s, nsat]:** We evaluate Cosmos Policy in the widely used RoboCasa benchmark consisting of 24 kitchen manipulation tasks, following the protocol outlined in the RoboCasa paper [1] and used in several prior works [2, 3, 4, 5]. For each task, we evaluate success rate over 50 trials across 5 evaluation scenes with different floor plans/styles, and we compute average success rate across all 24 tasks over 3 random seeds. The evaluation only consists of **unseen object instances**, and 2 of 5 scenes per task include **styles never seen in the training data**. Cosmos Policy achieves a **state-of-the-art average success rate of 67.1%** while requiring significantly fewer training demonstrations than other works (50 versus $>$300), as shown in the table below (Table 2 in the revised submission). This result highlights both effective **generalization** as well as **data efficiency**. Note that several works [1, 2, 3, 5, 10] show clear scaling over the number of demonstrations: i.e., for a given method, success rate with 300 demonstrations surpasses success rates with 100, 50, and 30. For prior methods, we report the higher success rates at larger data scales to highlight the relative data efficiency of Cosmos Policy.

|   | # Training Demos per Task | Average Success Rate (24 Tasks) |
|---|--|---|
| GR00T-N1 [2]  |  300  | 49.6% |
|  UVA [6]   |  50 |  50.0%  |
|  DP-VLA [4]  |  3000 | 57.3% |
| GR00T-N1 + DreamGen [5] | 300 (+10000 synthetic) | 57.6% |
| GR00T-N1 + DUST [7] | 300 | 58.5% |
| UWM [8] | 1000 | 60.8% |
| $\pi_0$ [9] |  300 | 62.5% |
| GR00T-N1.5 [2] | 300 | 64.1% |
| Video Policy [10] | 300 | 66.0% |
| FLARE [3] | 300 | 66.4% |
| GR00T-N1.5 + HAMLET [11] | 300 | 66.4% |
| **Cosmos Policy (ours)** | **50** | **67.1%** |


* **Additional ablation experiments in RoboCasa [nsat]:** We also add a fine-grained ablation analysis that provides more insights about the effects of different components in the Cosmos Policy design, including the joint training objectives (policy + world model + value function optimization with split training batches) and auxiliary targets during policy training (predicting future state and value alongside actions). See Appendix A.4.1 and Table 5 for the discussions and results. In summary, combining actions and future state prediction has the most crucial effect on policy performance (44.4% to 62.5% success rate), though adding other elements such as auxiliary value prediction for policy training and balanced splitting of training batches (into policy / world model / value function training samples) also leads to small increases in success rate (1 to 2 point increases in success rate for each component).

## Additional Details and Clarifications (Implementation, Training, Evaluation)
Further, we have added several new sections to the Appendix, with comprehensive details on implementation, training, and evaluation. We hope that the additional discussions and visualizations improve clarity (for example, on the latent injection mechanism). In addition, we have added a Reproducibility Statement at the end of the main text as well; upon publication, we will release project code, model checkpoints, and datasets needed to reproduce our training and evaluation runs.


## Discussions with Specific Reviewers
To address specific comments that the reviewers have made, we reply directly to each reviewer.


## References

[1] S. Nasiriany et al. Robocasa: Large-scale simulation of everyday tasks for generalist robots. RSS 2024.

[2] J. Bjorck et al. Gr00t n1: An open foundation model for generalist humanoid robots. arXiv:2503.14734, 2025.

[3] R. Zheng et al. Flare: Robot learning with implicit world modeling. CoRL 2025.

[4] B. Han et al. A dual process vla: Efficient robotic manipulation leveraging vlm. arXiv:2410.15549, 2024.

[5] J. Jang et al. Dreamgen: Unlocking generalization in robot learning through video world models. CoRL 2025.

[6] S. Li et al. Unified video action model. RSS 2025.

[7] J. Won et al. Dual-stream diffusion for world-model augmented vision-language-action model. arXiv:2510.27607, 2025.

[8] C. Zhu et al. Unified world models: Coupling video and action diffusion for pretraining on large robotic datasets. RSS 2025.

[9] K. Black et al. pi0: A vision-language-action flow model for general robot control. RSS 2025.

[10] J. Liang et al. Video generators are robot policies.arXiv:2508.00795, 2025.

[11] M. Koo et al. Hamlet: Switch your vision-language-action model into a history-aware policy.
arXiv:2510.00695, 2025.

---

### Meta-Review · Area_Chair_TLsD · 2026-01-10

**Summary:**

Most reviewers were positive about this paper, appreciating the interesting idea and the strong results. The raised concerns were about missing details and analyses, in particular, the latency issue. The authors successfully addressed most of the concerns, and the overall ratings remained positive. AC thus recommends accepting this paper.

**Reviewer Concerns:**

The rebuttal clarified missing details and also provided additional analyses and strong results. The issue of demonstrating performance under difference architectures and model sizes (reviewr i745) remains, but AC finds this is not critical.

**Reviewer Scores:**

Reviewer nsat would have retained the original rating of 8.
Reviewer i7f5 retained the original rating of 8 after discussion.
Reviewer nGa7 would have changed the score from 6 to 8.
Reviewer pF5s would have changed the score from 4 to 6.

---

### Decision · Program_Chairs · 2026-01-26

Accept (Poster)